# External Disturbances Rejection for Vector Field Sensors in Attitude and Heading Reference Systems

**DOI:** 10.3390/mi11090803

**Published:** 2020-08-25

**Authors:** Yongjun Wang, Zhi Li, Xiang Li

**Affiliations:** 1Guangxi Key Laboratory of Automatic Detecting Technology and Instruments, Guilin University of Electronic Technology, Guilin 541004, China; dongwang@guat.edu.cn; 2Key Laboratory of Unmanned Aerial Vehicle Telemetry, Guilin University of Aerospace Technology, Guilin 541004, China; cclizhi@guat.edu.cn

**Keywords:** accelerometers, attitude and heading reference systems (AHRS), Kalman filter, magnetic disturbance, magnetometers, sensor fusion

## Abstract

The attitude and heading reference system (AHRS), which consists of tri-axial magnetometer, accelerometer, and gyroscope, has been widely adopted for three-dimensional attitude determination in recent years. It provides an economical means of passive navigation that only relies on gravity and geomagnetic fields. However, despite the advantages of small size, low cost, and low power, the magnetometer and accelerometer are susceptible to external disturbances, such as the magnetic interference from nearby ferromagnetic objects and current-carrying conductors, as well as the motional acceleration of the carrier. To eliminate such disturbances, a vector-based parallel structure is introduced for the attitude filter design, which can avoid the mutual interference between gravity and geomagnetic vectors. Meanwhile, an approach to estimate and compensate the external disturbances in real time for magnetometer and accelerometer is also presented. Compared with existing designs, the proposed filter architecture and external disturbance rejection algorithm can feasibly and effectively cooperate with mainstream data fusion techniques, including complementary filter and Kalman filter. According to experiment results, in the case that large and persistent external disturbances exist, the proposed method can improve the accuracy and robustness of attitude estimation, and it outperforms the existing methods such as switching filter and adaptive filter. Furthermore, through the experiments, the critical role of fading factor in handling the external disturbance is revealed.

## 1. Introduction

Attitude and heading reference systems (AHRS) can provide three-dimensional attitude information [1,2,3,4,5,6], and they are widely used in unmanned aerial vehicles (UAV), mobile robots, motion tracking, etc. A typical sensor configuration in AHRS consists of a tri-axial magnetometer, a tri-axial accelerometer, as well as a tri-axial gyroscope. The combination of magnetometer, accelerometer, and gyroscope is also referred to as a MARG sensor [7,8,9], which has the advantages of ultra-low size, cost, and power.

Three-dimensional attitude estimation in AHRS mainly relies on two natural vector fields, namely gravity and geomagnetic fields [1,10]. The former points to the center of the Earth and can be measured by the accelerometer, while the latter points to the magnetic north and can be measured by the magnetometer. Besides the above two vectors, the gyroscope can provide the angular velocity of the carrier that the AHRS is attached to, and it can help to augment dynamic attitude accuracy.

In recent decades, theories and techniques for MARG sensor-based AHRS have been well developed. The scalar checking (or ellipsoid fitting) method [1,11,12,13,14,15] and the multi-position calibration [16,17] can help to compensate MARG sensor errors, including the bias, scale factors, misalignments, etc. Meanwhile, various solutions for MARG sensor fusion can also be found in the voluminous literature, such as the Kalman filter (KF) [1,2,3,7] and complementary filter (CF) [6,8,18,19,20].

Still, the complicated and volatile environments can bring in external disturbances for MARG-based AHRS. For instance, the motional acceleration of carrier (also known as external acceleration) will be vectoral added to gravity, and hence the calculation of pitch and roll angles will be affected [1,2,3]. On the other hand, external magnetic interference can lead to significant heading error [21], since geomagnetic field intensity is very low. Such external disturbances cannot be thoroughly solved by the above-mentioned error calibration and data fusion methods, since most of these methods are mainly based on time-invariant models. Hence, specific approaches are necessary to improve the accuracy and robustness of MARG-based AHRS against temporary external disturbances.

The existing external disturbances rejection algorithms can be categorized into two major types [5]: The threshold-based [1,3] and the model-based methods [2].

The threshold-based methods use one or more criteria (e.g., whether the norm of gravity or geomagnetic vector has notable deviation) to detect the external disturbance. They are also called switching filters, and can be further divided into ‘hard-switching’ and ‘soft-switching’ ones. The former directly excludes the vector field sensor from the filter once it is affected by external disturbance [3,21,22], while the latter gradually reduces the weight of the affected sensor for data fusion when the disturbance increases [1,9,23,24,25,26,27,28,29,30].

On the other hand, the model-based methods use certain model to estimate the external disturbance in real time, and then subtract it from the measurement of the corresponding sensor [2,31,32,33,34,35,36,37].

Nevertheless, the problem of external disturbance rejection still needs further studies. It is noteworthy that the existing approaches were mostly designed for specific attitude filter architecture, i.e., for most cases, an approach designed for KF cannot be directly applied to CF due to the difference in architecture, and vice versa. For instance, the method in [37] estimates the disturbances outside the KF, but it also modifies the covariance matrix in KF, and thus it is not fit for CF. Moreover, since it works outside but is not independent of the attitude filter, its mechanism needs more discussion.

In this paper, a more versatile approach is discussed and evaluated to estimate and compensate the external disturbances for vector field sensors (i.e., the accelerometer and magnetometer) in AHRS. This algorithm can cooperate with different types of attitude filters, and thus it is broadly applicable for MARG-based attitude determination.

The rest of this paper is organized as follows. The existing algorithms will be briefly surveyed in Section 2, and then the versatile algorithm for disturbance rejection will be introduced in Section 3. After that, experimental evaluation and concluding remarks of the proposed algorithm will be presented in Section 4 and Section 5, respectively.

## 2. Previous Works

### 2.1. MARG Sensor Error Modeling

As stated above, the MARG sensor in AHRS are used to measure three different vectors, namely the gravity vector g, the geomagnetic vector h, and the angular velocity ω. The measurement model of MARG sensor can be written as (1), which includes various error sources [1,11,12,13,14,15,16,17].
(1)$$\left\{ \begin{array}{l} v_{acc}=C_{sf,\;acc}C_{ma,acc}C_{no,acc}\left(g+f_{a}\right)+b_{acc}+\varepsilon_{acc}\\ v_{mag}=C_{sf,mag}C_{ma,mag}C_{no,mag}C_{si}\left(h+b_{hi}\right)+b_{mag}+\varepsilon_{mag}\\ v_{gyr}=C_{sf,gyr}C_{ma,gyr}C_{no,gyr}\omega +b_{gyr}+\delta b_{gyr}+\varepsilon_{gyr} \end{array}\right.$$

In Equation (1), the subscripts ‘acc’, ‘mag’, and ’gyr’ indicate the parameters corresponding to accelerometer, magnetometer, and gyroscope, respectively. The 3 × 1 vector v, b, and ε denote the sensor outputs, biases, and noise terms, respectively. Moreover, the 3 × 3 matrices $C_{sf}$, $C_{ma}$, and $C_{no}$ stand for the scale factors, misalignment, and non-orthogonality, respectively. Meanwhile, there are several noticeable issues in Equation (1) for each sensor.

According to Equation (1), the accelerometer is sensitive to both the specific force $f_{a}$ (defined as the non-gravitational force per unit mass) and gravity g. In the case that accelerometer is used to measure the motional acceleration [38], the measurand is $f_{a}$, while gravity g can be viewed as the disturbance. However, in AHRS, g is needed for attitude estimation, and $f_{a}$ plays the role of external disturbance.

For the magnetometer, the magnetic interferences can be categorized into hard-iron and soft-iron disturbances. The soft-iron disturbance is proportional to the external magnetic field, and it is described by the matrix $C_{si}$ in Equation (1). On the other hand, the hard-iron disturbance usually comes from the permanent magnetism of nearby ferromagnetic materials, and it is described by the 3 × 1 vector $b_{hi}$ in Equation (1). In AHRS, the geomagnetic vector h is used for attitude estimation, and $b_{hi}$ is the source of external disturbance.

For the gyroscope, $\delta b_{gyr}$ denotes the drift of its bias $b_{gyr}$, and it should be properly handled in attitude estimation algorithm to avoid accumulative error.

The measurement model in Equation (1) can be rewritten as Equation (2), in which $g^{*}$, $h^{*}$, and $\omega^{*}$ stand for the measurements of g, h, and ω, respectively. Moreover, the 3 × 3 matrix K and 3 × 1 vector b with corresponding subscripts indicate the deterministic sensor errors. Moreover, $\delta b_{gyr}$ is simplified to δb, while $d_{a}$ and $d_{m}$ indicate the undetermined external disturbances for the accelerometer and magnetometer, respectively.
(2)$$\{\begin{array}{l} g^{*}=K_{acc}\cdot g+b_{acc}+d_{a}+\varepsilon_{acc}\\ h^{*}=K_{mag}\cdot h+b_{mag}+d_{m}+\varepsilon_{mag}\\ \omega^{*}=K_{gyr}\cdot \omega +b_{gyr}+\delta b+\varepsilon_{gyr} \end{array}$$

The commonly used calibration methods for MARG sensor (such as the ellipsoid fitting method and multi-position method) can determine K and b, but most of them presume that both K and b are time-invariant. In other words, such calibrations are based on linear time-invariant (LTI) error models, and thus they are insufficient to compensate δb, $d_{a}$, and $d_{m}$.

In the following discussion, it is presumed that the time-invariant error terms of MARG sensor have already been calibrated appropriately.

### 2.2. MARG Sensor Data Fusion

As stated above, KF and CF are the two major types of MARG sensor fusion techniques.

There are two essential parts in KF, namely the state transition (or prediction) and state correction steps. As can be seen in Figure 1a, when KF is used for attitude estimation, in each time step, the 3D attitude Θ is predicted according to $\omega^{*}$, and then the prediction is corrected according to $g^{*}$ and $h^{*}$. The superscripts ‘−’ and ‘+’ indicate the a priori and a posteriori estimation, respectively.

Either the prediction or correction step may involve nonlinear mathematics, and thus nonlinear KFs had been proposed, e.g., the extended Kalman filter (EKF), unscented Kalman filter (UKF) [2], and cubature Kalman filter (CKF) [39,40]. Meanwhile, since there are various representations of 3D attitude Θ, such as Euler angles (i.e., heading, pitch, and roll), quaternion, direction cosine matrix (DCM), and even the vectors g and h [41,42], they also lead to numerous attitude filters.

On the other hand, complementary filters perform data fusion in frequency domain, as shown in Figure 1b,c. In Figure 1b, H(s) stands for a high-pass transfer function that can help to eliminate the gyro bias drift, and TRIAD refers to the tri-axial attitude determination algorithm [43]. Figure 1c shows another widely used implementation of CF that first presented in [18].

Hereafter, it is assumed that the noise term ε and the gyro bias drift δb can be properly handled by the attitude filter (i.e., KF or CF), and thus the only problem left is to handle the external disturbances $d_{a}$ and $d_{m}$.

### 2.3. Switching Filters

Different criteria can be used to detect the disturbances $d_{a}$ and/or $d_{m}$:-The norm of $g^{*}$ (or $h^{*}$): it is the most widely used criterion, due to its high convenience and feasibility.-The deviation between $\hat{g}$ and $g^{*}$ (or $\hat{h}$ and $h^{*}$): it is another frequently used criterion, which can be easily acquired from the residual (or innovation) in the state correction step of KF. However, it may be affected by the filter’s instability.-The angular velocity: it can be used to detect rotational acceleration (especially the centripetal acceleration) [28], but it is not always reliable since the turning radius is usually unknown.-The magnetic dip (i.e., the angle between h and the horizontal plane): it can be used to detect $d_{m}$ [21,27], but with the requisite that the pitch and roll angles are accurately known.

Besides the variety of criteria, there are also different ways to implement hard-switching and soft-switching filters:(a)Tune the filter gain (or the weighting coefficient, cut-off frequency, etc.) directly [8,27].(b)Enlarge the measurement covariance of $g^{*}$ and/or $h^{*}$, to adjust the filter gain indirectly [1,23,24,25]. This is only fit for KF.(c)Tune the filter according to more complex switching logic, such as the hidden Markov model in [28] and the state machine in [30].

### 2.4. External Disturbances Estimation

Model-based disturbance rejection method tries to extract the external disturbances from raw measurement, rather than discarding it. Therefore, it can probably preserve more information than the threshold-based methods.

To estimate the external disturbance, it can be included in the state variables of attitude filter (specifically the KF) [31,32]. However, it will increase the dimension of KF, as well as the computational burden.

Alternatively, the external disturbance can be estimated outside the attitude filter, to avoid state augmentation [33,34,35,36,37]. Hereinafter, these two approaches are referred to as ‘interior’ estimator and ‘exterior’ estimator, respectively.

The principle of exterior estimator can be summarized as follows.(1)Assuming that the a posteriori estimation of $d_{a}$ and $d_{m}$ at the (*k* − 1)th time step are already available (denoted as ${\hat{d}}_{a,k-1}^{+}$ and ${\hat{d}}_{m,k-1}^{+}$), the a priori estimations at the *k*th time step (denoted as ${\hat{d}}_{a,k}^{-}$ and ${\hat{d}}_{m,k}^{-}$) are calculated according to Equation (3). Please note that the coefficient $c_{a}$ and $c_{m}$ are both between 0 and 1.
(3)$$\{\begin{array}{l} {\hat{d}}_{a,k}^{-}=c_{a}\cdot {\hat{d}}_{a,k-1}^{+}\\ {\hat{d}}_{m,k}^{-}=c_{m}\cdot {\hat{d}}_{m,k-1}^{+} \end{array}$$(2)Using ${\hat{d}}_{a,k}^{-}$ and ${\hat{d}}_{m,k}^{-}$, the measurements $g_{k}^{*}$ and $h_{k}^{*}$ are corrected according to Equation (4).
(4)$$\{\begin{array}{l} g_{k}^{c}=g_{k}^{*}-{\hat{d}}_{a,k}^{-}\\ h_{k}^{c}=h_{k}^{*}-{\hat{d}}_{m,k}^{-} \end{array}$$(3)Once the a posteriori estimation ${\hat{g}}_{k}^{+}$ and ${\hat{h}}_{k}^{+}$ are provided by KF, the estimations of $d_{a}$ and $d_{m}$ can be updated according to Equation (5).
(5)$$\{\begin{array}{l} {\hat{d}}_{a,k}^{+}=g_{k}^{*}-{\hat{g}}_{k}^{+}\\ {\hat{d}}_{m,k}^{+}=h_{k}^{*}-{\hat{h}}_{k}^{+} \end{array}$$

The exterior estimator described by Equations (3)–(5) has been proven to be effective in [33,34,35,36,37], but it can only work with KF. Meanwhile, it still has two major problems that need further explanation.

The first problem is the interpretation of Equation (3), which was called the first-order auto-regressive process [36] or Markov chain process [37] that driven by low-pass filtered white noise [33,34,35]. However, in most cases, the external disturbances $d_{a}$ and $d_{m}$ are unpredictable, i.e., there is no sufficient a priori knowledge about their arising and changes over time. Therefore, Equation (3) is by no means rigorous, and it is certainly not driven by white noise (as presumed in [33,34,35,36,37]).

The second problem is the measurement covariance in KF when using the above exterior estimator. Theoretically, if the covariance of ${\hat{d}}_{a,k}^{-}$ is $R_{a}$, while the covariance of $g_{k}^{*}$ is $R_{g}$, the covariance of $g_{k}^{c}$ should be $\left(R_{g}+R_{a}\right)$. Unfortunately, $R_{a}$ remains unknown in most cases due to the lack of a priori knowledge. In [33,34,35,36,37], $R_{a}$ was calculated according to $d_{a}$ itself. For instance, $R_{a}$ can be simply defined as a scalar matrix in Equation (6) [33,37]:(6)$$R_{a,k}=\frac{1}{3}{\left|{\hat{d}}_{a,k}^{-}\right|}^{2}=\frac{1}{3}c_{a}^{2}{\left|{\hat{d}}_{a,k-1}^{+}\right|}^{2}$$

Alternatively, $R_{a}$ can be calculated by a windowed smoothing approach, as in Equation (7) [34]:(7)$$R_{a,k}=\frac{1}{m}c_{a}^{2}\overset{m}{\underset{i=1}{\sum }}{\hat{d}}_{a,k-i}^{+}{\left({\hat{d}}_{a,k-i}^{+}\right)}^{T}$$

Obviously, Equations (6) and (7) will enlarge the measurement covariance of $g_{k}^{c}$ once the external disturbance arises, but they are not rigorous either.

In the following section, the versatility of the exterior estimation algorithm will be expanded, and a different interpretation of Equation (3) will be given. Furthermore, it will be proved that the above modification of measurement covariance are unnecessary (and can even bring in negative effects) by experiments in Section 4.

## 3. Novel Attitude Filter Design

### 3.1. Attitude Filter Architecture

Figure 2 shows a universal architecture for attitude estimation as well as external disturbances rejection. This architecture can cooperate with all variants of KF and CF, since the estimation and compensation of either $d_{a}$ or $d_{m}$ is performed outside the filter. Moreover, this architecture incorporates several features that can help to simplify the structure and/or enhance the robustness.

First, the 3D attitude information is preserved in g and h, instead of the commonly used quaternion or DCM. Thus, there is no need to calculate the quaternion or DCM according to g and h, or vice versa. Once g and h have been estimated, the heading, pitch, and roll angles (denoted as ψ, θ, and φ, respectively) can be solved by TRIAD algorithm [43]. This feature is known as the vector-based or sensor-based design [41,42].

Secondly, since the filter is vector-based, it is possible to use two sub-filters to estimate g and h separately. This is called the parallel filter [37], which can avoid the mutual interference between g and h, and thus help to restrain the impact of external disturbances. Meanwhile, it is also beneficial for reducing the dimension of each sub-filter.

Last but not least, the external disturbances $d_{a}$ and $d_{m}$ are estimated outside the attitude filter, i.e., it is an exterior estimator of $d_{a}$ and $d_{m}$, as introduced in Section 2.4. Thus, it is possible to use different algorithms in the filter, including EKF, UKF, CKF, and CF.

### 3.2. External Disturbances Estimation

Figure 2 also shows the algorithm for external disturbances estimation, in which the superscripts ‘−’ and ‘+’ indicate the a priori and a posteriori estimations, respectively.

In each time step, the proposed attitude estimation and external disturbances rejection algorithm can be summarized as follows:Step 1: Predict the disturbances ${\hat{d}}_{a,k}^{-}$ and ${\hat{d}}_{m,k}^{-}$. This step is the same as (3).Step 2: Subtract ${\hat{d}}_{a,k}^{-}$ and ${\hat{d}}_{m,k}^{-}$ from the corresponding measurements (i.e., $g_{k}^{*}$ and $h_{k}^{*}$) to get the corrected vectors (denoted as $g_{k}^{c}$ and $h_{k}^{c}$). This step is the same as (4).Step 3: Get the estimated vectors ${\hat{g}}_{k}$ and ${\hat{h}}_{k}$ according to $g_{k}^{c}$, $h_{k}^{c}$, and $\omega_{k}^{*}$ through a certain algorithm (KF or CF).Step 4: Update the estimated disturbances ${\hat{d}}_{a,k}^{+}$ and ${\hat{d}}_{m,k}^{+}$ according to ${\hat{g}}_{k}$ and ${\hat{h}}_{k}$, i.e., ${\hat{d}}_{a,k}^{+}=g_{k}^{*}-{\hat{g}}_{k}$ and ${\hat{d}}_{m,k}^{+}=h_{k}^{*}-{\hat{h}}_{k}$. Then k=k+1 and go to Step 1.

### 3.3. Specific Implementations

The above algorithm can cooperate with KF and CF, as presented below.

#### 3.3.1. The Proposed Method with Kalman Filter (Abbreviated as PM-KF Hereafter)

PM-KF Step 1: Get a priori estimation according to Equation (8), in which the state transition matrix is calculated as $F_{k}=I_{3\times 3}-\Delta t\cdot {\left[\omega_{k}^{*}\right]}_{\times }$. The matrix ${\left[\omega_{k}^{*}\right]}_{\times }$ satisfies ${\left[\omega_{k}^{*}\right]}_{\times }u=\omega_{k}^{*}\times u$ for any 3 × 1 vector u, and Δt is the sampling period.
(8)$$\{\begin{array}{l} {\hat{g}}_{k}^{-}=F_{k}{\hat{g}}_{k-1}^{+}\\ {\hat{h}}_{k}^{-}=F_{k}{\hat{h}}_{k-1}^{+} \end{array}$$PM-KF Step 2: Calculate a priori covariance matrices according to (9). The process covariance matrices are defined as $Q_{g}={\left|g\right|}^{2}\sigma_{\omega }^{2}I_{3\times 3}$ and $Q_{h}={\left|h\right|}^{2}\sigma_{\omega }^{2}I_{3\times 3}$, with $\sigma_{\omega }$ denoting the noise of gyroscope.
(9)$$\{\begin{array}{l} P_{k,g}^{-}=F_{k}P_{k-1,g}^{+}F_{k}^{T}+Q_{g}\\ P_{k,h}^{-}=F_{k}P_{k-1,h}^{+}F_{k}^{T}+Q_{h} \end{array}$$PM-KF Step 3: Get a posteriori estimation according to Equations (10)–(12). The measurement covariance matrices are defined as $R_{g}=\sigma_{g}I_{3\times 3}$ and $R_{h}=\sigma_{h}I_{3\times 3}$, with $\sigma_{g}$ and $\sigma_{h}$ denoting the noise of accelerometer and magnetometer, respectively.
(10)$$\{\begin{array}{l} K_{k,g}=P_{k,g}^{-}{\left(P_{k,g}^{-}+R_{g}\right)}^{-1}\\ K_{k,h}=P_{k,h}^{-}{\left(P_{k,h}^{-}+R_{h}\right)}^{-1} \end{array}$$
(11)$$\{\begin{array}{l} g_{k}^{c}=g_{k}^{*}-{\hat{d}}_{a,k}^{-}\\ h_{k}^{c}=h_{k}^{*}-{\hat{d}}_{m,k}^{-} \end{array}$$
(12)$$\{\begin{array}{l} {\hat{g}}_{k}^{+}={\hat{g}}_{k}^{-}+K_{k,g}\left(g_{k}^{c}-{\hat{g}}_{k}^{-}\right)\\ {\hat{h}}_{k}^{+}={\hat{h}}_{k}^{-}+K_{k,g}\left(h_{k}^{c}-{\hat{h}}_{k}^{-}\right) \end{array}$$PM-KF Step 4: Calculate a posteriori covariance matrix according to Equation (13).
(13)$$\{\begin{array}{l} P_{k,g}^{+}=\left(I_{3\times 3}-K_{k,g}\right)P_{k,g}^{-}\\ P_{k,h}^{+}=\left(I_{3\times 3}-K_{k,h}\right)P_{k,h}^{-} \end{array}$$

Then k=k+1 and go to PM-KF Step 1.

#### 3.3.2. The Proposed Method with Complementary Filter (Abbreviated as PM-CF Hereafter)

PM-CF Step 1: Correct the angular velocity according to Equation (14).
(14)$$\{\begin{array}{l} \omega_{k,g}^{c}=\omega_{k}^{*}-K_{g}\cdot \omega_{k-1,g}^{err}\\ \omega_{k,h}^{c}=\omega_{k}^{*}-K_{h}\cdot \omega_{k-1,h}^{err} \end{array}$$PM-CF Step 2: Update gravity and geomagnetic vectors using to the corrected angular velocity, as described by Equations (15)–(17).
(15)$$\{\begin{array}{l} g_{k,p}={\hat{g}}_{k-1}+{\hat{g}}_{k-1}\times \omega_{k-1,g}^{c}\\ h_{k,p}={\hat{h}}_{k-1}+{\hat{h}}_{k-1}\times \omega_{k-1,h}^{c} \end{array}$$
(16)$$\{\begin{array}{l} g_{k,c}={\hat{g}}_{k-1}+g_{k,p}\times \omega_{k,g}^{c}\\ h_{k,c}={\hat{h}}_{k-1}+h_{k,p}\times \omega_{k,h}^{c} \end{array}$$
(17)$$\{\begin{array}{l} {\hat{g}}_{k}=\left(g_{k,c}+g_{k,p}\right)/2\\ {\hat{h}}_{k}=\left(h_{k,c}+h_{k,p}\right)/2 \end{array}$$PM-CF Step 3: Compensate external disturbances and calculate equivalent angular velocity errors, as described by Equations (18) and (19). Δt denotes the sampling period.
(18)$$\{\begin{array}{l} g_{k}^{c}=g_{k}^{*}-{\hat{d}}_{a,k}^{-}\\ h_{k}^{c}=h_{k}^{*}-{\hat{d}}_{m,k}^{-} \end{array}$$
(19)$$\{\begin{array}{l} \omega_{k,g}^{err}=\frac{{\hat{g}}_{k}\times g_{k}^{c}}{\left|{\hat{g}}_{k}\right|\left|g_{k}^{c}\right|\cdot \Delta t}\\ \omega_{k,h}^{err}=\frac{{\hat{h}}_{k}\times h_{k}^{c}}{\left|{\hat{h}}_{k}\right|\left|h_{k}^{c}\right|\cdot \Delta t} \end{array}$$

Then k=k+1 and go to PM-CF Step 1.

### 3.4. Remarks

As can be seen in the above implementations, the proposed parallel vector-based architecture can greatly simplify the design of attitude filter, especially the KF. There is no need to use UKF, CKF, or other nonlinear version of KF, since both the process and measurement models are concise and straightforward. Moreover, the proposed external disturbance rejection method seems to be a direct extension of Equations (3)–(5). However, it can be interpreted in a different way as follows.

As pointed out in the above section, the external disturbances $d_{a}$ and $d_{m}$ are unpredictable in most cases. As a matter of fact, the essence of Equation (3) is the assumption that either $d_{a}$ or $d_{m}$ will not have drastic change during a single sampling period, and thus the last a posteriori estimation can be used to correct the sensor measurement at the present time step. Obviously, this approach is inexact, and the actual role of the coefficients $c_{a}$ or $c_{m}$ is to make the estimation error tend to decrease. In other words, $c_{a}$ and $c_{m}$ are actually the fading factors that help to avoid divergence. Nonetheless, if $c_{a}$ (or $c_{m}$) is too close to zero, it will inevitably weaken the compensation of $d_{a}$ (or $d_{m}$). The effects of these two fading factors will be evaluated in the following section.

To find a general solution for not only the KF, it is unnecessary to modify the covariance of $g_{k}^{c}$ and/or $h_{k}^{c}$. In fact, the proposed algorithm makes no change to the attitude filter itself, and thus it is an actual exterior estimator of external disturbances. Moreover, it can be seen in the following experiments that the modification of the measurement covariance is not always helpful to improve the performance.

## 4. Experiments

### 4.1. Basic Settings

The proposed algorithm is evaluated on an AHRS module, which is based on a monolithic MARG sensor MPU9250 and working at the sampling rate of 20 Hz. Meanwhile, a single-axis rate table is used to provide heading and angular rate references with 0.0001° resolution. The AHRS module and rate table, as well as the hardware installation, are shown in Figure 3.

The magnetometer and accelerometer in MPU9250 are calibrated using a calibration scheme based on dual inner products, which can be viewed as an improved ellipsoid fitting method and was detailed in [44]. Moreover, the gyroscope in MPU9250 is calibrated using a cross product-based algorithm, which was presented in [45].

After the above calibration, raw data are then acquired according to the flowchart in Figure 4. It can be seen that the AHRS experiences high-speed stop-and-go rotations, which can generate considerable centripetal acceleration. Meanwhile, it also suffers artificial magnetic interference that caused by ferromagnetic objects.

Three different algorithms are used for attitude estimation, including PM-KF and PM-CK that introduced in Section 3.3, as well as the parallel Kalman filter in [37] (abbreviated as PA-KF). The main difference between PM-KF and PA-KF is that the latter has adaptive measurement covariance, as described by Equation (6).

In both PM-KF and PA-KF, the sensor noise covariances are $\sigma_{h}=0.5\;\mu T$, $\sigma_{g}=0.1\;m/s^{2}$, and $\sigma_{\omega }=0.005\;\mathrm{rad}/s$, respectively. On the other hand, the coefficients of PM-CF are set to $K_{g}=K_{h}=0.3$.

### 4.2. Experiment Results

Since each of the above algorithms outputs the estimations of g and h, their performance is evaluated in terms of the directional errors of g and h, i.e., the angle between the estimated vector and the true vector. At the *k*th time step, if the estimated vector is ${\hat{g}}_{k}$ (or ${\hat{h}}_{k}$), while the true vector is $g_{r,k}$ (or $h_{r,k}$), the directional error can be calculated as $\delta_{g,k}=\cos^{-1}\left[\left({\hat{g}}_{k}\cdot g_{r,k}\right)/\left(∥{\hat{g}}_{k}∥∥g_{r,k}∥\right)\right]$ (or $\delta_{h,k}=\cos^{-1}\left[\left({\hat{h}}_{k}\cdot h_{r,k}\right)/\left(∥{\hat{h}}_{k}∥∥h_{r,k}∥\right)\right]$). Moreover, both fading factors $c_{a}$ and $c_{m}$ are increased from 0 to 2 with the increment of 0.01, so as to evaluate their impacts. Figure 5 shows the maximum error and root mean square error (RMSE) of each algorithm versus the fading factors.

As can be seen in Figure 5, the fading factors $c_{a}$ and $c_{m}$ have significant impacts on the performance of each algorithm. It is clear that all the algorithms will diverge once $c_{a}$ or $c_{m}$ is greater than 1. Meanwhile, in Figure 5, the optimal fading factors for each algorithm (i.e., the fading factors that lead to the best performance) can be found, as listed in Table 1. It is noteworthy that the optimal fading factors of PA-KF are quite small, especially the optimal value of $c_{a}$ is only around 0.05 (but coincides with [33] and [37]). As mentioned in Section 3.4, such small fading factors will weaken the compensation of external disturbances.

To further demonstrate the performance of each algorithm with its optimum fading factors, the estimated vectors g and h, the estimated disturbances $d_{a}$ and $d_{m}$, as well as the corresponding 3D attitude outputs (heading ψ, pitch θ, and roll φ) are all plotted in Figure 6 and Figure 7.

In Figure 6a,b, the centripetal acceleration caused by rapid rotations can be spotted, and it results in the fluctuation and divergence in the output of PA-KF in Figure 6d,e. On the contrary, the outputs of PM-KF and PM-CF remains stable when the centripetal acceleration exists. This is consistent to the results in Table 1, i.e., both PM-KF and PM-CF have better results than PA-KF in compensating the large and lasting external acceleration.

Nevertheless, it can also be noticed in Table 1 that PM-KF and PM-CF do not outperform PA-KF when compensating magnetic interference. More details of the magnetic interference and its compensation by different algorithms are shown in Figure 7.

## 5. Discussion

The above experiment results have proven the feasibility and effectiveness of the proposed method, including the vector-based parallel architecture and the external disturbance rejection algorithm. Still, there are some noteworthy issues.

First, the proposed method (including PM-KF and PM-CF) outperforms PA-KF when dealing with the centripetal acceleration, but shows no superiority in handling the magnetic interference. As shown in Figure 6, the raw data from accelerometer contain large and long-term centripetal acceleration, and the proposed method is more adaptive to such hostile conditions. On the contrary, PA-KF works better against momentary and moderate disturbances, as shown in Figure 7.

The main reason for the above phenomenon lies in the data fusion process of MARG sensor. Once the external disturbance arises, the measurement of accelerometer and/or magnetometer becomes unreliable, and thus the attitude estimation algorithm can only rely on the integration of angular velocity. However, inevitable bias drift of gyroscope results in a dilemma that the attitude filter will either diverge due to accumulated error, or still use the distorted measurement of accelerometer and/or magnetometer. In a word, any solution for disturbance rejection is essentially a trade-off between the gyro drift and the external disturbances.

Compared to the switching filters that completely discard the unreliable measurements, both PA-KF and the proposed method partially retain the measurements of accelerometer and magnetometer even when they contain external disturbances. However, PA-KF assigns more weight to the gyroscope by enlarging the measurement covariance of accelerometer and/or magnetometer. Consequently, PA-KF will be more significantly affected by the gyro drift when the disturbance is large and lasting.

Another key issue is the fading factor $c_{a}$ and $c_{m}$, which can greatly impact the compensating of external disturbance. As discussed in Section 3.4, a fading factor between 0 and 1 can help to decrease the estimation error, and it is proven by the fact that the optimal fading factors of all the algorithms are less than 1 in Table 1. However, if the fading factor is too small, it will evidently weaken the compensation of external disturbance, and it is also proven by the results listed in Table 1.

Compared to PM-KF and PM-CF, the optimal fading factors of PA-KF are much closer to 0. Since PA-KF is more significantly affected by the gyro drift due to its adjustment to the measurement covariance, it needs a smaller fading factor to alleviate this problem. Unfortunately, such a small fading factor will definitely weaken the compensation effect, and it results in the poor performance of PA-KF when handling large and lasting centripetal acceleration.

Finally, it is worth mentioning again that the main advantage of the proposed method is not only its better performance against strong and persistent disturbances, but also its flexibility to cooperate with commonly used sensor fusion algorithms (including but not limited to KF and CF).

## 6. Conclusions

In this paper, a versatile external disturbance rejection approach is presented, along with a vector-based parallel architecture for 3D attitude estimation. It is proven by experiments that the proposed filter architecture and disturbance rejection approach can work well with KF and CF in the presence of external acceleration and magnetic interference.

The proposed method provides a feasible and generally applicable solution for MARG-based 3D attitude estimation. Nonetheless, the problem of external disturbance rejection for accelerometer and magnetometer is far from completely solved. As stated in the above section, all solutions for disturbance rejection in MARG-based AHRS, either the existing approaches or the proposed method, are no more than trade-offs between the gyro drift and the external disturbance. Since the significant role of the fading factor is revealed, a possible direction for future exploring is the adaptive adjustment of such fading factors.

## Figures and Tables

**Figure 1 micromachines-11-00803-f001:**
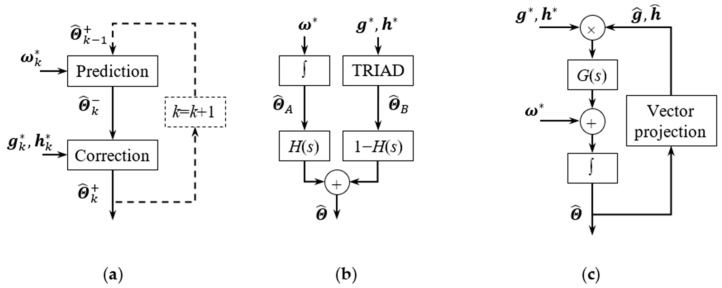
Block diagram of Kalman filter and complementary filter for 3D attitude estimation. (**a**) Kalman filter. (**b**) Classic complementary filter. (**c**) Mahony’s complementary filter.

**Figure 2 micromachines-11-00803-f002:**
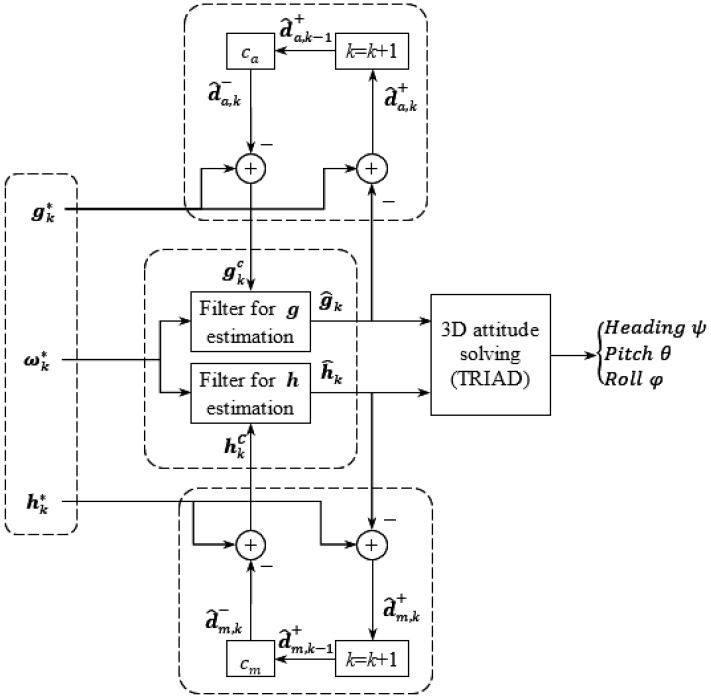
Block diagram of vector-based parallel attitude filter with external disturbances rejection.

**Figure 3 micromachines-11-00803-f003:**
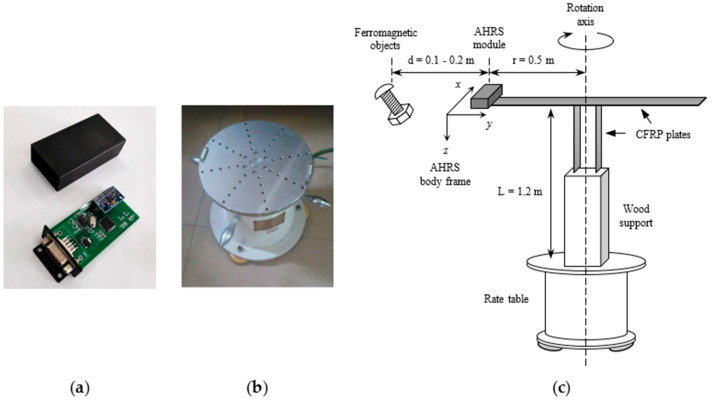
Experimental apparatus. (**a**) AHRS module. (**b**) Single-axis rate table. (**c**) Hardware installation (not to scale).

**Figure 4 micromachines-11-00803-f004:**
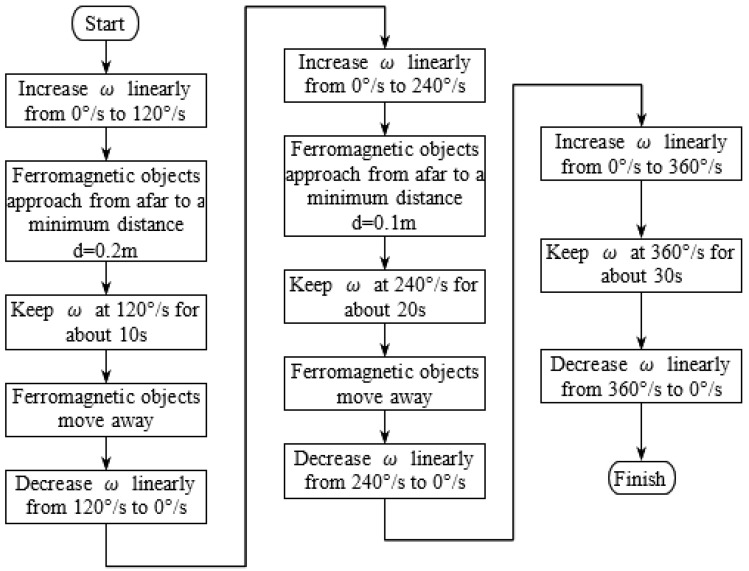
Flowchart for raw data acquisition.

**Figure 5 micromachines-11-00803-f005:**
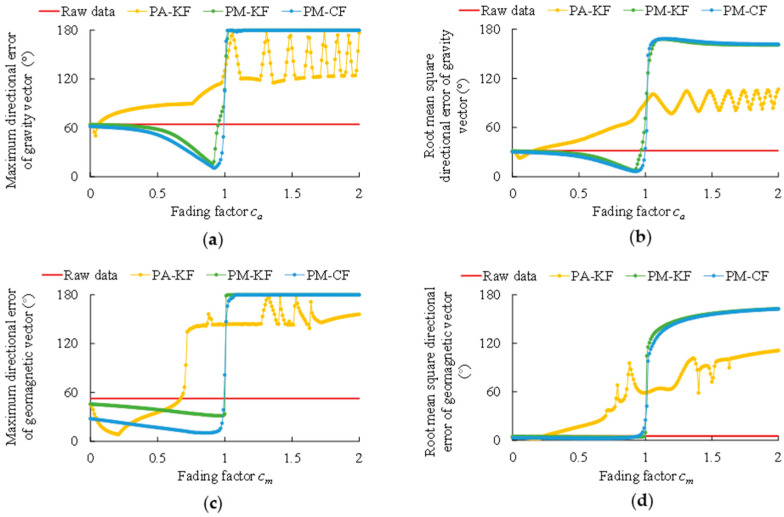
Performance of each algorithm versus fading factors, in terms of the directional errors of gravity vector g and geomagnetic vector h. (**a**) Maximum error of g. (**b**) RMSE of g. (**c**) Maximum error of h. (**d**) RMSE of h.

**Figure 6 micromachines-11-00803-f006:**
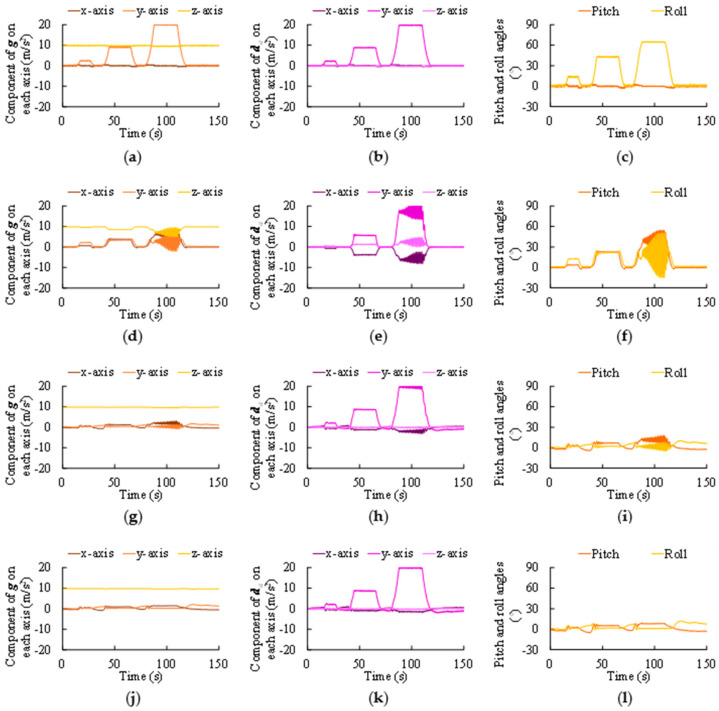
Estimation of gravity vector g and disturbance $d_{a}$, as well as the corresponding pitch (θ) and roll (φ) angles. (**a**) g measurements, with uncompensated $d_{a}$. (**b**) Theoretical value of $d_{a}$. (**c**) θ and φ calculated from uncompensated g measurements. (**d**) g estimated by PA-KF with $c_{a}=0.05$. (**e**) $d_{a}$ estimated by PA-KF. (**f**) θ and φ calculated from PA-KF outputs. (**g**) g estimated by PM-KF with $c_{a}=0.92$. (**h**) $d_{a}$ estimated by PM-KF. (**i**) *θ* and *φ* calculated from PM-KF outputs. (**j**) g estimated by PM-CF with $c_{a}=0.93$. (**k**) $d_{a}$ estimated by PM-CF. (**l**) θ and φ calculated from PM-CF outputs.

**Figure 7 micromachines-11-00803-f007:**
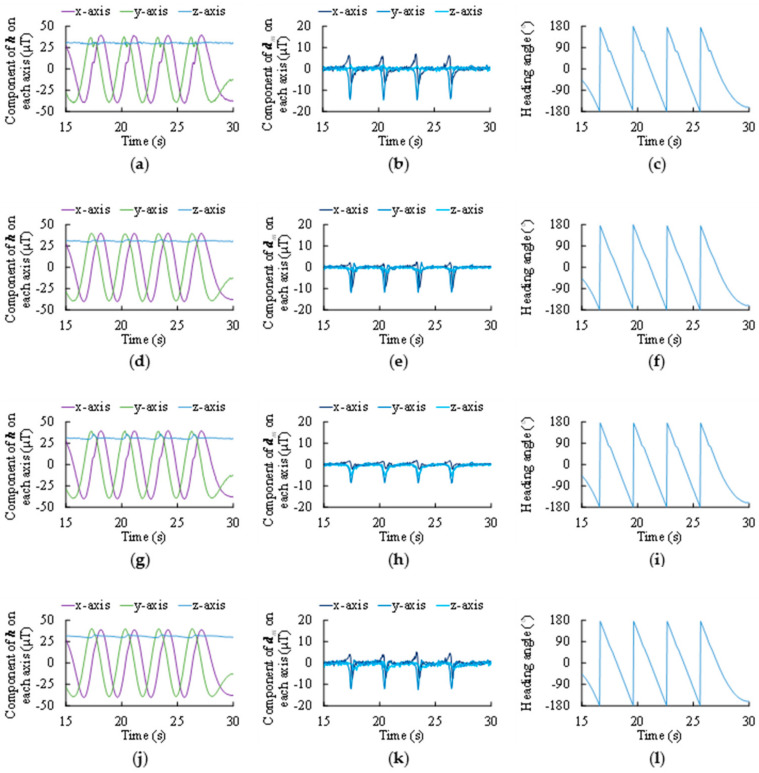
Estimation of geomagnetic vector h and disturbance $d_{m}$, as well as the corresponding heading (ψ) angle. (**a**) h measurements, with uncompensated $d_{m}$. (**b**) Theoretical value of $d_{m}$. (**c**) ψ calculated from uncompensated h measurements. (**d**) h estimated by PA-KF with $c_{m}$ = 0.20. (**e**) $d_{m}$ estimated by PA-KF. (**f**) ψ calculated from PA-KF outputs. (**g**) h estimated by PM-KF with $c_{m}$ = 0.83. (**h**) $d_{m}$ estimated by PM-KF. (**i**) ψ calculated from PM-KF outputs. (**j**) h estimated by PM-CF with $c_{m}$ = 0.64. (**k**) $d_{m}$ estimated by PM-CF. (**l**) ψ calculated from PM-CF outputs.

**Table 1 micromachines-11-00803-t001:** The best performance of each algorithm.

Algorithm		Raw Data	PA-KF	PM-KF	PM-CF
Gravity vector	Maximum error/$c_{a}$	64.36°/−	50.34°/0.04	15.63°/0.91	10.58°/0.92
	RMSE/$c_{a}$	31.72°/−	22.97°/0.05	7.41°/0.92	6.46°/0.93
Geomagnetic vector	Maximum error/$c_{m}$	52.75°/−	8.24°/0.21	31.31°/0.96	10.26°/0.87
	RMSE/$c_{m}$	5.24°/−	1.76°/0.20	3.82°/0.83	2.18°/0.64

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
