# Peer review of "External Disturbances Rejection for Vector Field Sensors in Attitude and Heading Reference Systems"

_micromachines, 2020, doi:10.3390/mi11090803_

Round 1

Reviewer 1 Report

The proposed manuscript presents a method for the estimation and compensation of external disturbances for vector field sensors.

The paper is well organized, but it has multiple deficiencies.
The main problem is that many parts are not described in enough detail. The style of English used in the text should be also improved.

Regarding the proposed model and method, the authors did not describe the error types for the sensors in enough detail and the applied sensor models are not correct.
In the case of the magnetic sensor, the applied error model is also too simple. Magnetometers are affected by hard- and soft-iron effects (these are caused by the external disturbances) beside the systematic errors (scale factor, bias and misalignment error) (see the two references below). In Eq. (1), dm (which is obviously the hard-iron effect) should be multiplied by Kmag, since it is affected by these errors, and soft-iron effects should be also added.

G. Secer, and B. Barshan, Improvements in deterministic error modeling and calibration of inertial sensors and magnetometers. Sensors and Actuators A: Physical 247: 522-538, 2016.
V. Renaudin, M. H. Afzal, and G. Lachapelle, Complete triaxis magnetometer calibration in the magnetic domain. Journal of Sensors 2010: 967245, 2010.

In the case of the accelerometer and the gyroscope, the used models are also not correct, since the external disturbances should be multiplied by Kacc and Kgyro.

Although, the utilized error models are not correct, it does not have an effect on the proposed method, since the authors assume that the time-invariant errors are calibrated. It should be also pointed out that the soft-iron effects are ignored in the proposed model.

The models are also validated with experimental measurements, and are compared with standard methods.

Detailed suggestions:
-The manuscript contains some errors in spelling and grammar which need to be corrected.
-Do not use pronouns "we" and "our" in the text, use passive voice instead.
-The style of English in the whole paper should be improved to be more "scientific".
-The sensor error models for the different sensor types should be described in more detail as mentioned above.
-It should be pointed out that the external disturbance in the case of the accelerometer (da) is the gravitational force, which generates an acceleration with 1g magnitude.
-The term "always vertical downwards" (page 1, line 38) should be changed to: "points to the center of the Earth".
-The titles of Figure 1 and Figure 4 should be fixed.
-In Figure 2 the meaning of "and keep" should be described.
-The authors assume in the model that the time-invariant errors are calibrated in the case of all sensors. Please, describe in the text with which method was the used sensor unit calibrated.
-The generation of magnetic disturbances during the data acquisition should be described in more detail.
-It should be described for which angle are the errors calculated in Figure 5. If it is only for one angle, the other two should be also examined.
-In Figure 5, the size of the graphs (c) and (d) should be the same as (a) and (b), since it makes easier to analyze the results.
-A figure showing the error in the angles per axis should be added.
-A figure of the estimated external disturbances should be also added, and it would be also interesting to see the magnitude of the disturbances.

Author Response

We appreciate your suggestions for enhancing the paper quality. Please see the attached file.

Reviewer 2 Report

the authors presented a versatile external disturbance rejection approach, along with a vector-based parallel architecture for 3D attitude estimation. They also proved by experiments that the proposed filter architecture and disturbance rejection approach can work well with KF and CF in the presence of external acceleration and magnetic interference. The proposed method provides a feasible and generally applicable solution for MARG-based 3D attitude estimation. Nonetheless, the problem of external disturbance rejection for accelerometer and magnetometer is far from completely solved. As stated in the above section, all solutions for disturbance rejection in MARG-based AHRS, either the existing approaches or the proposed method, are no more than trade-offs between the gyro drift and the external disturbance.

Paper contains a contribution that may be of interest for readers. Results seems to be correct.

The paper could be improved by extension of the list of references and discussion of the novelty of the presented results comparing to the existing state. Citation to many papers of Prof. Roman Czyba should be considered.

Summing up, I recommend this article for acceptance to publish in Journal.

Author Response

Thank you very much for your review and recommendation. As you suggest, more related articles have been added to the reference list, including some inspiring articles from Prof. Roman Czyba (numbered as [6], [20], and [38] in the revised paper), so as to better present the state of art.

Round 2

Reviewer 1 Report

The authors made the required changes and properly answered questions of the reviewers. So, I suggest accepting the paper.